# RTMF-Net: A Dual-Modal Feature-Aware Fusion Network for Dense Forest Object Detection

**DOI:** 10.3390/s25185631

**Published:** 2025-09-10

**Authors:** Xiaotan Wei, Zhensong Li, Yutong Wang, Shiliang Zhu

**Affiliations:** 1Key Laboratory of the Ministry of Education for Optoelectronic Measurement Technology and Instrument, Academy of Smart IC and Network, Beijing Information Science and Technology University, Beijing 102206, China; 2023020554@bistu.edu.cn (X.W.); w18515656301@163.com (Y.W.); 2State Key Laboratory of Information Engineering in Surveying, Mapping and Remote Sensing, Wuhan University, Wuhan 430079, China; shiliangzhu@whu.edu.cn

**Keywords:** RGB-TIR, object detection, multimodal, lightweight, feature fusion

## Abstract

Multimodal remote sensing object detection has gained increasing attention due to its ability to leverage complementary information from different sensing modalities, particularly visible (RGB) and thermal infrared (TIR) imagery. However, existing methods typically depend on deep, computationally intensive backbones and complex fusion strategies, limiting their suitability for real-time applications. To address these challenges, we propose a lightweight and efficient detection framework named RGB-TIR Multimodal Fusion Network (RTMF-Net), which introduces innovations in both the backbone architecture and fusion mechanism. Specifically, RTMF-Net adopts a dual-stream structure with modality-specific enhancement modules tailored for the characteristics of RGB and TIR data. The visible-light branch integrates a Convolutional Enhancement Fusion Block (CEFBlock) to improve multi-scale semantic representation with low computational overhead, while the thermal branch employs a Dual-Laplacian Enhancement Block (DLEBlock) to enhance frequency-domain structural features and weak texture cues. To further improve cross-modal feature interaction, a Weighted Denoising Fusion Module is designed, incorporating an Enhanced Fusion Attention (EFA) attention mechanism that adaptively suppresses redundant information and emphasizes salient object regions. Additionally, a Shape-Aware Intersection over Union (SA-IoU) loss function is proposed to improve localization robustness by introducing an aspect ratio penalty into the traditional IoU metric. Extensive experiments conducted on the ODinMJ and LLVIP multimodal datasets demonstrate that RTMF-Net achieves competitive performance, with mean Average Precision (mAP) scores of 98.7% and 95.7%, respectively, while maintaining a lightweight structure of only 4.3M parameters and 11.6 GFLOPs. These results confirm the effectiveness of RTMF-Net in achieving a favorable balance between accuracy and efficiency, making it well-suited for real-time remote sensing applications.

## 1. Introduction

### 1.1. Background

Object detection in remote sensing imagery plays a vital role in various critical applications, including military reconnaissance [1], emergency response [2], urban management [3], and resource monitoring. In recent years, the continuous growth in high-resolution remote sensing data [4], together with the rapid advancement of deep learning techniques, has significantly boosted the performance of object detection methods based on convolutional neural networks (CNNs). Among them, optical remote sensing images have become the mainstream data source for object detection due to their rich texture details and spatial structural information. However, in practical scenarios, complex natural environments, such as mountainous and forested regions, are often accompanied by vegetation occlusion, terrain variations, and artificial camouflage, which greatly reduce the visibility of ground objects and increase the difficulty of recognition. Moreover, optical sensors are highly sensitive to lighting conditions, and their imaging quality tends to degrade significantly under low-light environments, strong glare, or adverse weather such as fog and cloud cover [5,6], thereby affecting the stability and robustness of detection systems, as illustrated in Figure 1. These challenges are especially pronounced in remote sensing imagery: on the one hand, the high-altitude imaging results in a substantial reduction in object size; on the other hand, complex backgrounds and multiple sources of interference further complicate object detection tasks [7], revealing the limitations of single-modality optical approaches in terms of both detection accuracy and generalization ability.

To address the perceptual limitations of optical modalities, thermal infrared (TIR) imagery has been introduced into remote sensing object detection tasks. By capturing the thermal radiation emitted by objects, TIR images exhibit strong robustness to environmental variations and can reliably acquire object features under nighttime or low-illumination conditions. Nevertheless, due to their relatively low spatial resolution and significant loss of fine details, TIR images often suffer from degraded thermal contrast, especially in scenarios with high ambient temperatures or minimal temperature differences between the object and background. This results in blurred object boundaries and indistinct contours, which severely hinder detection performance. Consequently, relying on a single modality is insufficient to achieve accurate and robust object perception in complex remote sensing environments. There is an urgent need to explore multimodal data fusion strategies that can leverage complementary information across modalities to enhance detection robustness and accuracy.

To address the limitations of single-modality imagery under complex environmental conditions, an increasing number of studies have focused on exploring multi-modal data fusion strategies in remote sensing [8,9], aiming to overcome the expressive constraints of unimodal representations. For instance, CMFR [10] introduces cross-modality correction mechanisms in both spatial and channel dimensions to effectively suppress modality-specific noise. MIN [11] adopts a concise fusion strategy to mitigate perceptual interference in optical images caused by cloud occlusion. MoCG [12] leverages global complementary features from optical images to enhance the robustness of synthetic aperture radar (SAR) data under challenging conditions, such as speckle noise and perspective distortion. These methods demonstrate promising capabilities in improving fusion quality and generalization performance across modalities.

Additionally, the Multispectral Detection Transformer (DETR) [13] employs two separate backbone networks for feature extraction from RGB and infrared (IR) modalities while introducing a spectral switching mechanism to alleviate extraction bias. ICAFusion [14], based on a Transformer architecture, constructs a Dual-Modality Feature Fusion (DMFF) module to capture long-range dependencies and deeply integrate complementary information across modalities. Although these approaches exhibit strong performance in modeling multimodal interactions, their highly complex structures often incur substantial computational costs, which limits their applicability in edge computing or resource-constrained scenarios. To balance efficiency and performance, SuperYOLO [15] concatenates RGB and IR images at the channel level and utilizes a unified backbone network to improve inference speed and training efficiency. It further introduces an auxiliary super-resolution (SR) branch and a compact symmetric multimodal fusion module (MF) to enhance small object detection. However, its fusion strategy lacks sufficient modeling of intermodal interactions, resulting in underutilization of complementary features. This may lead to reduced robustness under modality interference or adversarial conditions. CLGNet [16] focuses on correcting spatial misalignment between modalities and suppressing redundant information yet still falls short in dynamic modeling and coordinated utilization of multimodal features.

In summary, there remains a critical need to develop a lightweight and computationally efficient multimodal object detection network that supports effective modality interaction. Such a design would significantly enhance the perception capability and practical applicability of remote sensing systems in complex real-world environments.

### 1.2. Main Contributions

To enhance the detection performance of small objects in complex remote sensing scenes, this paper proposes a low-complexity multimodal object detection network, termed RTMF-Net, which jointly optimizes the modality fusion strategy and feature extraction mechanisms to achieve a better balance between accuracy and efficiency. Specifically, a weighted denoising fusion module is introduced in the fusion stage, where an adaptive weighting mechanism is employed to suppress redundant and noisy information, thereby improving the complementarity and discriminability of multimodal features. In the visible modality branch, a computation-efficient feature extraction module (CEFBlock) is designed to enhance semantic modeling capability while maintaining low computational overhead. In the infrared branch, a Dual-Laplacian Enhancement Block (DLEBlock) is constructed to extract multi-scale frequency-domain structural features, improving texture representation in thermal infrared imagery. Additionally, an adaptive fusion unit is integrated into the neck network to dynamically adjust the fusion ratio between semantic and detailed features at different levels, achieving more precise feature integration. To address the challenges posed by large object scale variation and significant viewpoint changes in remote sensing imagery, a novel Shape-Aware IoU (SA-IoU) loss function is further proposed. This loss function incorporates geometric shape modeling, which significantly improves the robustness and accuracy of object localization.

The major contributions of this paper are summarized as follows:1.We propose a multimodal object detection network named RTMF-Net, specifically designed for scenarios with occluded objects. Existing methods often fail to fully exploit complementary information from optical (RGB) and thermal infrared (TIR) modalities, especially under occlusion or varying object scales. To address this gap, RTMF-Net combines modality-specific backbone enhancements, a weighted denoising fusion module, and a three-branch detection head, which together enable more effective feature extraction, adaptive fusion, and multi-scale detection. This combination is necessary to fully leverage multimodal information and improve detection robustness.2.To strengthen the modeling capability of multimodal features, we, respectively, integrate the CEFBlock and DLEBlock into the optical and thermal infrared branches. The CEFBlock improves the representation ability of the optical backbone for multi-scale objects, while the DLEBlock enhances the thermal infrared backbone’s ability to extract weak texture and frequency-domain information. These enhancements facilitate more effective representation of both shared and complementary features between modalities at the feature extraction stage.3.We further design a weighted denoising fusion module to improve the discriminative power during the feature fusion stage. This module adaptively integrates feature maps from the optical and infrared backbones and incorporates a custom-designed EFA mechanism, which effectively suppresses redundant information and highlights salient object regions, thereby enhancing the overall detection performance.4.To address the performance degradation caused by viewpoint variations in the ODinMJ [17] dataset, we propose an improved overlap evaluation metric named SA-IoU. This metric introduces an aspect ratio penalty term into the conventional IoU calculation, thereby improving the model’s robustness in scenarios with significant object scale changes and pose variations, and enhancing both localization precision and detection accuracy.

The remainder of this paper is organized as follows: Section 2 reviews object detection methods and RGB-TIR fusion-based detection approaches. Section 3 introduces the proposed modules in detail. Section 4 presents the datasets and experimental results. In Section 5, we systematically evaluate the effectiveness of each module in the model, identify its current limitations, and provide potential optimization directions for future research. Section 6 concludes the paper and outlines future research directions.

## 2. Related Works

### 2.1. Object Detection Algorithms

CNN-based object detection methods can be broadly categorized into two types: two-stage detectors and one-stage detectors. Two-stage methods, such as R-CNN [18], Fast R-CNN [19], Faster R-CNN [20], and Mask R-CNN [21], first generate candidate regions via selective search or region proposal networks (RPN), followed by feature extraction, classification, and regression on each region. These methods typically achieve high detection accuracy but suffer from relatively high computational complexity and slower inference speed, limiting their applicability in real-time scenarios. In contrast, one-stage detectors, including SSD [22] and the You Only Look Once (YOLO) series [23,24,25,26,27,28,29], directly predict object classes and bounding box coordinates on feature maps extracted from the entire image, streamlining the detection pipeline and significantly improving inference speed, making them more suitable for real-time applications.

The YOLO series has become a prominent research focus due to its end-to-end detection architecture and extremely high inference efficiency. While YOLO demonstrates strong speed performance, challenges remain in handling small objects, occlusions, and complex backgrounds, which can affect robustness and localization accuracy. Transformer-based methods, such as DETR (DEtection TRansformer) [30], complement CNN-based detectors by modeling global relationships across the image. DETR formulates object detection as a direct set prediction problem and eliminates the need for hand-crafted components such as anchor boxes and NMS, enabling competitive accuracy.

To address the inherent limitations of YOLO, various improvements have been proposed, including multi-scale feature fusion mechanisms to enhance perception of objects at different scales [31,32,33]; attention mechanisms to focus on critical regions [34,35]; anchor optimization or anchor-free strategies to improve detection accuracy and generalization [36]; and advanced loss functions such as IoU-based losses and Focal Loss to enhance training efficiency and performance. Additionally, combining data augmentation, diverse sampling strategies, lightweight network architectures, and pruning techniques can further accelerate inference while maintaining robustness and accuracy in complex scenarios. These advancements provide a solid foundation for object detection in high-complexity applications, such as remote sensing imagery.

### 2.2. RGB-TIR Fusion Detection Algorithms

In complex and dynamic environments, single-modality imagery often fails to meet the demand for high-precision object detection. While optical images provide rich texture and color features, their effectiveness deteriorates significantly under uneven illumination, strong direct light, or low light/nighttime conditions. TIR images capture objects’ thermal radiation distributions directly and are invariant to ambient lighting, enabling clear visualization of object contours and thermal characteristics in night, haze, and heavy occlusion scenarios. Moreover, compared to optical images, TIR imagery inherently excels in detecting concealed or camouflaged objects by supplying complementary semantic information, thereby enhancing overall detection performance. Against this backdrop, fusing RGB and TIR modalities can compensate for the limitations of each modality and substantially improve the adaptability and robustness of detection systems in complex scenes.

In recent years, researchers have proposed various dual-branch network architectures to address multimodal fusion by separately extracting features from different modalities and then enabling effective interaction and complementarity during the fusion stage. Such approaches not only fully preserve the unique advantages of each modality but also significantly improve detection accuracy and robustness through the design of reasonable fusion mechanisms. Among related works, Fei et al. [37] proposed ACDF-YOLO, a multimodal fusion detection framework based on YOLOv5. This model integrates an efficient channel attention module (ESA) and a cross-modal difference module (CDM) to strengthen both intra-modal feature extraction and inter-modal complementary information fusion, demonstrating the effectiveness of deep fusion strategies in multimodal remote sensing object detection. Zhang et al. [38] proposed a dual-branch multimodal diffusion model for cloud removal in remote sensing images, which effectively improves the recovery of fine-grained targets and robustness in complex scenarios through cross-modal attention fusion and adaptive image prediction strategies. Meng et al. [39] developed a lightweight RGB-IR object detection network, FQDNet, which achieves multimodal feature fusion via a channel exchange SCDown module (CSSB) and a spatial-channel attention fusion module (SCAFM). It further enhances multi-scale feature utilization through a dynamic weighted four-head detector (DWQH), balancing detection accuracy and computational efficiency. Zhang and Sun et al. [40] proposed the lightweight multispectral object detection network GMD-YOLO, which strengthens modal complementarity by fusing multi-layer features, improving detection accuracy while maintaining high efficiency. Cao et al. [41] introduced SSCD-YOLO, a semi-supervised cross-domain pedestrian detection model that reduces domain discrepancies between infrared and visible images, effectively enhancing detection performance and model robustness in low-light environments.

Inspired by these works, this paper proposes RTMF-Net, which designs different backbone networks tailored to each modality for feature extraction, incorporates a weighted denoising fusion module to suppress redundant interference and enhance information complementarity, and adds an adaptive fusion node in the neck to optimize the balance between detail and semantic features, thereby improving detection performance.

## 3. Method

### 3.1. Overall Architecture

Figure 2 presents the overall architecture of the proposed RTMF-Net. Object detection in occluded environments, this model constructs a dual-modality fusion detection network based on RGB and TIR images. RTMF-Net is built upon the lightweight YOLOv8n framework and primarily consists of five core components: an optical feature extraction backbone, a thermal-infrared feature extraction backbone, a multimodal feature fusion module, a feature integration neck, and the detection head.

In the optical branch, the CEFBlock is introduced, comprising standard convolutional layers and an improved C2f_FEM module, which effectively enhances the spatial representation capabilities of visible images across varying viewpoints. The thermal infrared branch incorporates the DLEBlock, composed of frequency-aware LPConv and C2f modules, which strengthens the extraction of thermal infrared texture and edge features. To address the multimodal information fusion challenge, a Weighted Denoising Fusion Module is proposed, which adaptively weights visible and infrared features to effectively suppress background noise and enhance the discriminability of fused features. The detection head retains the original three-scale detection structure of YOLOv8n, ensuring robust detection performance across objects of varying scales.

Overall, RTMF-Net achieves significant improvements in detection accuracy under complex scenarios while maintaining low parameter count and computational complexity. The following sections will provide a detailed discussion of the design principles and implementation details of each module.

### 3.2. Convolutional Enhanced Feature Block

To address challenges in remote sensing images such as large variations in object scale and complex viewing angles, this work designs the CEFBlock structure within the optical feature extraction backbone, as illustrated in Figure 2. This structure consists of two standard convolutional layers and an improved C2f_FEM module. Unlike the traditional C2f module, C2f_FEM replaces part of the Bottleneck units with a specially designed Feature Enhancement Module (FEM) to better capture spatial deformations caused by viewpoint shifts, terrain undulations, and sensor angles. These factors often lead to geometric distortions in remote sensing imagery, where objects may appear stretched, rotated, or skewed compared to their canonical shapes. By incorporating asymmetric convolutions and multi-directional feature modeling, the CEFBlock enhances robustness to such distortions and improves the representation of objects under varying perspectives and appearances.

In response, a three-branch Feature Enhancement Module is designed to strengthen the network’s ability to model spatial structures, multi-scale details, and directional features, thereby improving the diversity and robustness of feature representations. As illustrated in Figure 2, this module employs a multi-branch parallel architecture that integrates concepts such as asymmetric convolutions, multi-directional modeling, and local perception. Specifically, it consists of the following three parallel branches:

Branch 0: Utilizes 1 × 1 pointwise convolution to compress or expand input feature channels, reducing redundancy and computational cost while providing a unified dimension for subsequent multi-branch fusion.

Branch 1: Implements an asymmetric convolutional structure, initially adjusting channel dimensions via 1 × 1 convolution, followed by sequential 1 × 3 and 3 × 1 convolutions to extract spatial features along horizontal and vertical directions, effectively enhancing the module’s sensitivity and expressiveness toward directional features.

Branch 2: Incorporates shallow convolutions or lightweight residual units to supplement local details such as edges and textures, improving the module’s adaptability to small objects and complex backgrounds across different scenarios.

The outputs from the three branches are concatenated along the channel dimension for feature fusion, significantly enhancing the overall representational capacity. This design achieves improved feature abstraction while maintaining low computational overhead, facilitating more accurate object detection and recognition in remote sensing contexts.(1)$$W_{1}=f_{conv}^{1\times 1}(F)$$(2)$$W_{2}=f_{conv}^{3\times 1}\left\{f_{conv}^{1\times 3}\left[f_{conv}^{1\times 1}(F)\right]\right\}$$(3)$$W_{3}=f_{conv}^{3\times 3}\left[f_{conv}^{1\times 1}(F)\right]$$(4)$$Y=f_{conv}^{1\times 1}[cat(W_{1},W_{2},W_{3})]$$

In Equations (1)–(4), $f_{Conv}^{1\;\times \;1}$, $f_{Conv}^{3\;\times \;1}$, $f_{Conv}^{1\times 3}$, $f_{Conv}^{3\times 3}$ denote convolution operations with kernel sizes of 1 × 1, 3 × 1, 1 × 3, and 3 × 3, respectively. F represents the input feature map, $W_{1}$, $W_{2}$ and $W_{3}$ are the outputs of the three branches, and Y is the final output.

### 3.3. Dual-Domain Lightweight Extraction Block

The Pinwheel Convolution (PConv) [42] is a lightweight, structure-aware convolutional module designed to enhance fine-grained target perception through multi-directional feature modeling. Aimed at addressing the anisotropic thermal radiation, weak texture, and blurred edges commonly observed in TIR imagery, this module introduces directional receptive kernels (e.g., horizontal, vertical, and diagonal) to simulate a pinwheel-like convolutional pattern. This design effectively strengthens the network’s sensitivity to gradient changes across different orientations, expanding the spatial receptive field without significantly increasing computational cost. As a result, it improves feature extraction for low-contrast and weakly bounded small targets.

It is important to note that, in contrast to traditional infrared imagery formed by reflected external infrared light, thermal infrared images are generated based on the inherent thermal radiation of objects. Thus, TIR imagery primarily reflects the temperature distribution rather than surface texture. Due to this imaging mechanism, TIR images tend to exhibit low signal-to-noise ratios, weak texture information, and blurred edges—conditions that limit the responsiveness of traditional directional convolution in weak-boundary regions, thereby hindering the accurate detection and separation of small targets.

The first layer of PConv performs parallel convolutions, which can be formulated as follows:(5)$$X_{1}^{\left(h^{\prime },w^{\prime },c^{\prime }\right)}=SiLU\left(BN\left(X_{P(0,1,0,3)}^{\left(h_{1},w_{1},c_{1}\right)}⊗W_{1}^{\left(1,3,c^{\prime }\right)}\right)\right)$$(6)$$X_{2}^{\left(h^{\prime },w^{\prime },c^{\prime }\right)}=SiLU\left(BN\left(X_{P(0,3,0,1)}^{\left(h_{1},w_{1},c_{1}\right)}⊗W_{2}^{\left(3,1,c^{\prime }\right)}\right)\right)$$(7)$$X_{3}^{\left(h^{\prime },w^{\prime },c^{\prime }\right)}=SiLU\left(BN\left(X_{P(0,1,3,0)}^{\left(h_{1},w_{1},c_{1}\right)}⊗W_{3}^{\left(1,3,c^{\prime }\right)}\right)\right)$$(8)$$X_{4}^{\left(h^{\prime },w^{\prime },c^{\prime }\right)}=SiLU\left(BN\left(X_{P(3,0,1,0)}^{\left(h_{1},w_{1},c_{1}\right)}⊗W_{4}^{\left(3,1,c^{\prime }\right)}\right)\right)$$

Here, ⊗ denotes the convolution operation, $w_{1}^{{(1,3,c}^{\prime })}$ is a 1×3 convolution kernel with $c^{\prime }$ output channels. The padding parameter P (1,0,0,3) represents the number of padding pixels on the left, right, top, and bottom, respectively.

To further enhance the edge-awareness of PConv in TIR images, this study designs a Fixed Gradient Attention (FGA) module. The FGA module applies a fixed Laplacian operator to generate edge-response maps, which are then processed via depthwise separable convolutions along directional convolution paths. This design allows FGA to selectively emphasize regions with strong intensity variations while maintaining computational efficiency, as it introduces no additional trainable parameters. The rationale behind combining FGA with PConv is that PConv captures orientation-specific features, emphasizing gradients along certain directions, whereas FGA provides an isotropic response to intensity changes, highlighting edges regardless of orientation. By fusing these two mechanisms, the model benefits from both directional sensitivity and global edge amplification, effectively enhancing weak or low-contrast edges. The edge-aware features extracted by FGA are subsequently fused with the frequency-domain representations from PConv, further improving the model’s ability to perceive fine edge details and enhancing the detection of small targets in thermal infrared scenes.

As illustrated in Figure 3a presents the basic structure of the PConv module. Unlike standard convolution, this module performs asymmetric padding along four directions on the input tensor to construct a multi-directional, diffused receptive field, thereby enabling stronger orientation-aware modeling. Suppose the input tensor is $x\in R^{c_{1}\times H_{1}\times w_{1}}$, $c_{1}$, $H_{1}$, $w_{1}$ denote the number of channels, height, and width, respectively. To improve training stability and convergence, each convolution operation is followed by batch normalization (BN) and a Sigmoid Linear Unit (SiLU) activation function. Figure 3b shows the enhanced structure with Laplacian edge guidance. After obtaining multi-directional frequency responses via PConv, edge information is further emphasized through deep Laplacian convolution, significantly improving detection performance—particularly in handling weak boundaries and blurred targets within thermal infrared imagery.

### 3.4. Improve Fusion Module

To enhance the representational capacity of fused multi-source features while suppressing redundant noise information, a Fusion Noise Reduction Module is proposed, as illustrated in Figure 4. This module consists of two main stages:

Firstly, feature maps from the two backbone networks are integrated using a weighted fusion strategy, referred to as RTFusion. This strategy employs a learnable parameter α to model the relative importance of features from different modalities, resulting in an initial fused representation:(9)$$F_{\mathrm{fused}}=\alpha \cdot F_{1}+(1-\alpha )\cdot F_{2}$$

Here, $F_{1}$ and $F_{2}$ denote the input features from the two backbone networks, and α represents a learnable fusion weight parameter.

Subsequently, to further suppress noise and enhance the discriminability of target regions, the fused feature is fed into an enhanced attention mechanism, namely the Enhanced Fusion Attention (EFA) module. This mechanism integrates two complementary attention pathways: Modified Channel Attention (MCA) and Modified Spatial Attention (MSA). Specifically:

The channel attention branch utilizes global average pooling and max pooling to capture feature distribution statistics, which are then adaptively fused using learnable weights. A two-layer fully connected network is employed to model inter-channel dependencies, thereby emphasizing semantically relevant feature channels.

The spatial attention branch first compresses the channel dimension using a 1 × 1 convolution, followed by the extraction of local spatial context information. This is combined with both average-pooled and max-pooled feature maps to generate a spatial attention map that emphasizes target-relevant regions in the fused representation.

The attention maps obtained from the two branches are subsequently combined via a learnable normalized fusion weight vector $w=\left[w_{c},w_{s}\right]$, which satisfies the constraint $w_{c}+w_{s}=1$. The final attention map is computed through a linear combination as follows:(10)$$A=w_{c}\cdot A_{c}+w_{s}\cdot A_{s}$$
where $A_{c}$ and $A_{s}$ represent the outputs of the channel and spatial attention branches, respectively. The final output feature is obtained by performing element-wise multiplication between the fused feature map $F_{fused}$ and the aggregated attention map A:(11)$$F_{\mathrm{out}}=F_{\mathrm{fused}}⊗A$$

This fusion attention mechanism not only enhances the response intensity in target-relevant regions but also effectively suppresses background and redundant noise features, thereby providing a more discriminative semantic representation for downstream detection tasks.

### 3.5. Shape Aware IoU

IoU is one of the most widely used evaluation metrics in object detection tasks, serving to quantify the overlap between the predicted bounding box and the ground truth. Beyond its role in model performance evaluation during the testing phase, IoU also plays a critical role in key components of the training pipeline, such as positive/negative sample assignment and Non-Maximum Suppression (NMS). However, IoU suffers from performance degradation when the predicted and ground truth boxes do not overlap, resulting in decreased accuracy, particularly for objects with complex shapes.

To address this issue, YOLOv8 adopts the Generalized IoU (GIoU) as the loss function. Although GIoU demonstrates certain advantages in object detection tasks, it still has notable limitations. Firstly, when the predicted box is fully enclosed within the ground truth box but is spatially offset, GIoU exhibits slow numerical variation, leading to weak gradient signals and consequently hindering convergence during training. Secondly, the GIoU loss primarily emphasizes the positional relationship between bounding boxes, penalizing based on the area difference of the smallest enclosing box, while largely neglecting shape disparities—especially differences in aspect ratio. As a result, when two boxes are close in center location but differ significantly in shape, GIoU fails to effectively penalize this misalignment.

Furthermore, the penalty term in GIoU is relatively simplistic, relying solely on the area difference between the enclosing box and the union of the two boxes. This design lacks the granularity needed to capture discrepancies in shape and aspect ratio, potentially allowing predicted boxes with considerable shape deviations to receive insufficient penalization. These limitations constrain the overall performance of GIoU in object detection tasks.

To overcome these drawbacks, we propose a novel loss function, Shape-Aware IoU (SA-IoU), formulated as follows:(12)$$SA-IoU=IoU-\alpha \cdot Penalty_{GIoU}-\gamma \cdot Penalty_{shape}$$(13)$$IoU=\frac{Area_{inter}}{Area_{union}}$$(14)$$Penalty_{GIoU}=\frac{\left|C\right|-\left|B_{p}\cup B_{g}\right|}{\left|C\right|}$$

To address this issue, this paper proposes SA-IoU, which explicitly introduces an aspect ratio penalty term on top of the traditional IoU and GIoU formulations to more comprehensively measure the geometric discrepancy between the predicted box and the ground truth box. Specifically, the aspect ratio penalty term is defined as the absolute difference between the width-to-height ratios of the predicted and ground truth boxes.(15)$$Penalty_{shape}=\left|\frac{w_{P}}{h_{P}}-\frac{w_{g}}{h_{g}}\right|$$

Here, $w_{p}$ and $h_{p}$ denote the width and height of the predicted box, respectively, while $w_{g}$ and $h_{g}$ represent the width and height of the ground truth box. A larger value of this term indicates a greater shape discrepancy between the predicted and ground truth boxes, thereby imposing a stronger penalty on the model and encouraging it to adjust the predicted box shape more precisely.

By explicitly introducing the aspect ratio penalty term, the SA-IoU loss function becomes more sensitive to geometric shape discrepancies between predicted and ground truth boxes, addressing the limitation of traditional GIoU, which primarily focuses on positional relationships while ignoring shape differences. The inclusion of this term strengthens the gradient signal, enabling the model to optimize the geometric structure of the predicted boxes more effectively, thereby significantly improving both convergence speed and localization accuracy. Furthermore, this term helps prevent the generation of predicted boxes with extreme aspect ratios during training, enhancing overall training stability and robustness.

## 4. Experiments

### 4.1. Experimental Environment

The experiments are conducted on a Windows 11 operating system, equipped with an RTX 4090 GPU (24 GB). The software environment consists of Python 3.8, PyTorch 2.0.0, and CUDA 11.8. Under consistent hyperparameter settings, training, validation, and testing are performed. The training batch size is set to 32, the initial learning rate is 0.01, and the total number of epochs is 200. The Adam optimizer is employed, and except for the proposed modifications, the remaining hyperparameters follow those of the original YOLOv8 framework.

### 4.2. Dataset

The ODinMJ dataset is a multimodal dataset designed for human detection tasks in mountainous and forested environments. Released by Kunming University of Science and Technology and utilized in a Kaggle competition this dataset contains 23,075 pairs of aligned visible and thermal infrared images, each with a resolution of 640 × 640 pixels. All images are captured from UAV perspectives in complex natural environments, covering diverse vegetation types, terrain slopes, lighting conditions, and camouflage states. The dataset includes a total of 45,617 annotated bounding boxes for the “person” category, comprehensively reflecting variations in occlusion, camouflage, and posture. We conducted a statistical analysis of the ODinMJ dataset using the COCO evaluation metrics, categorizing objects into three groups: small, medium, and large. The detailed distribution is presented in Table 1 and Figure 5.

The LLVIP dataset [43] is a large-scale multimodal image dataset specifically con-structed for pedestrian detection tasks in low-light conditions. It aims to evaluate the robustness and cross-modal perception capability of detection models under challenging lighting environments. The dataset consists of 14,491 pairs of corresponding visible and infrared images, spanning a wide range of typical lighting scenarios from daytime to nighttime, and from bright to extremely dark conditions. Similarly, we performed a statistical analysis on the LLVIP dataset. The results are illustrated in Table 2 and Figure 6.

### 4.3. Evaluation Metrics

In evaluating the performance of object detection models, commonly used metrics include Precision (P), Recall (R), and mAP, which collectively quantify the detection capabilities of the model. Among these, mAP serves as a crucial metric reflecting the overall detection performance, calculated as the mean of Average Precision (AP) values across all categories. AP is defined as the area under the Precision-Recall curve plotted at varying confidence thresholds. Precision indicates the proportion of correctly identified targets among all detections made by the model, while Recall represents the model’s ability to capture all relevant targets. The formulas for these evaluation metrics are as follows:(16)$$P=\frac{T_{p}}{T_{p}+F_{P}}\times 100\%$$(17)$$R=\frac{T_{p}}{T_{p}+F_{N}}\times 100\%$$

Specifically, Precision represents the proportion of true positive samples among all samples predicted as positive, while Recall reflects the proportion of actual positive samples that are correctly identified. Here, TP denotes the number of true positives—samples correctly predicted as positive by the model; FP denotes false positives—samples incorrectly predicted as positive when they are actually negative; FN denotes false negatives—samples incorrectly predicted as negative when they are actually positive. The calculation formulas for AP and mAP are as follows:(18)$$AP=\int_{0}^{1}P\left(R\right)dR$$(19)$$mAP=\frac{1}{n}\sum_{i=1}^{n}AP_{i}$$

In addition to pursuing high detection performance, we also place great emphasis on model efficiency. Therefore, we adopt Giga Floating-Point Operations per Second (GFLOPs) and the number of parameters (Params) as key evaluation metrics. By considering these two indicators comprehensively, we can effectively assess the balance between performance and efficiency, providing crucial guidance for selecting suitable models in practical applications. GFLOPs and Params evaluate the computational resource requirements and model scale, while Inference Time (IT) serves as an important criterion for assessing real-time performance. Together, these metrics enable a holistic evaluation of the trade-off between model accuracy and efficiency.

### 4.4. Ablation Experiments

We first conducted an ablation study on the hyperparameter α in the SA-IoU loss function by evaluating five different parameter settings. The experimental results demonstrate that the model achieves optimal performance when α = 0.5 and (1 − α) = 0.5, with a mAP_50_ of 94.9% and a mAP_50–95_ of 59.5%. As shown in Table 3, although the precision and recall exhibit slight fluctuations under different settings, the configuration with α = 0.5 yields the most balanced performance across all metrics. This indicates that this specific value facilitates a better trade-off between the model’s classification capability and localization accuracy. In this section, the bolded values in all tables indicate the best results.

To evaluate the effectiveness of each proposed module, we conduct comprehensive ablation experiments on the ODinMJ dataset. The experimental results, as summarized in Table 4 and Table 5, include both detection performance and model complexity.

Experiment A serves as the baseline and adopts the improved dual-backbone YOLOv8n structure, achieving a precision of 95.4%, recall of 92.5%, mAP_50_ of 96.7%, and mAP_50–95_ of 69.9%.

In Experiment B, the LPConv module is introduced into the infrared branch to enhance the modeling of fine-grained texture information in thermal images. This leads to a notable increase in both detection precision and robust-ness, raising the mAP50 to 97.8% and mAP_50–95_ to 72.7%.

Experiment C replaces the standard C2f module in the visible-light branch with the proposed C2f_FEM, which strengthens multi-scale and shape-aware representations. As a result, the model achieves an mAP_50_ of 97.7% and mAP_50–95_ of 72.8%.

Experiment D incorporates the proposed Weighted Denoising Fusion Module during the feature fusion stage. By adaptively suppressing background noise and recalibrating modality contributions, the model further improves to an mAP_50_ of 97.9% and mAP_50–95_ of 73.1%.

Experiment E explores the use of SA-IoU as the training loss function to enhance shape-level discrimination. Although the loss modification does not alter the inference architecture, the network benefits from stronger shape alignment during training, resulting in an mAP_50_ of 97.1% and mAP_50–95_ of 71.1%.

Experiments F, G, and H progressively integrate the aforementioned modules to assess their combined effects. Experiment F, which fuses LPConv and C2f_FEM, achieves an mAP_50_ of 98.0% and mAP_50–95_ of 73.9%. Further inclusion of the Weighted Denoising Fusion Module in Experiment G improves these values to 98.3% and 75.0%, respectively. When all proposed modules are utilized in Experiment H, the model reaches the best performance with a precision of 97.4%, recall of 96.3%, mAP_50_ of 98.7%, and mAP_50–95_ of 75.3%, demonstrating the complementary nature and cumulative benefits of the proposed improvements.

In terms of model complexity, the baseline model in Experiment A comprises 4.3 million parameters, 12.3 GFLOPs, and an inference time of 0.5 ms. The introduction of LPConv in Experiment B does not increase the parameter count, while reducing GFLOPs slightly to 12.1 and adding minimal inference delay. The C2f_FEM module in Experiment C maintains comparable complexity. Experiment D shows a moderate increase in parameters and GFLOPs due to the addition of Weighted Denoising Fusion Module but remains within an efficient range. Since SA-IoU operates only during training, Experiment E maintains the same complexity as the baseline. The integration of multiple modules in Experiments F to H demonstrates that the proposed architecture maintains efficiency even in its most complete form. Specifically, Experiment H yields the highest accuracy while keeping the parameter count at 4.2 M, GFLOPs at 11.6, and inference time within 0.9 ms, validating the practicality of the overall design.

We also conducted ablation experiments on the LLVIP dataset to further validate the effectiveness of each proposed module. The evaluation results, summarized in terms of detection accuracy, are presented in Table 6 and Table 7.

Experiment A serves as the baseline, utilizing the dual-branch YOLOv8n without any of the proposed improvements. It achieves a Precision of 93.9%, Recall of 87.6%, mAP_50_ of 94.4%, and mAP_50–95_ of 57.8%, providing a reference point for subsequent enhancements.

Experiment B introduces the LPConv module into the infrared branch. This module enhances the modeling of fine-grained thermal textures, improving the network’s sensitivity to subtle heat signatures. Consequently, Recall increases to 89.2% and mAP_50–95_ reaches 60.1%, marking a 2.3% gain over the baseline.

Experiment C replaces the standard C2f module in the visible-light branch with the proposed C2f_FEM module, designed to better capture multi-scale and shape-aware features. This yields a Precision of 95.0%, Recall of 89.1%, and mAP_50–95_ of 60.3%.

Experiment D incorporates the Weighted Denoising Fusion Module during the multi-modal feature fusion phase. By adaptively emphasizing informative cross-modal features and suppressing background noise, the module improves Recall to 89.6% and mAP_50–95_ to 60.5%.

Experiment E replaces the conventional CIoU loss with the SA-IoU loss function to strengthen shape-level alignment and perception. While not changing the network architecture, this modification enhances training effectiveness, resulting in an mAP_50–95_ of 59.5%.

Experiment F combines LPConv and C2f_FEM, enhancing both thermal and visible branches simultaneously. This synergistic configuration improves Precision and Recall to 95.1% and 90.2%, respectively, with mAP_50–95_ reaching 61.2%.

Experiment G adds Weighted Denoising Fusion Module on top of the dual-branch enhancements, further boosting multi-modal interaction and yielding a Precision of 95.5%, Recall of 90.8%, and mAP_50–95_ of 61.4%.

Finally, Experiment H integrates all proposed modules—LPConv, C2f_FEM, Weighted Denoising Fusion Module, and SA-IoU—into the full architecture. This complete configuration achieves the best performance with a Precision of 95.3%, Recall of 91.3%, mAP_50_ of 95.7%, and mAP_50–95_ of 61.3%, demonstrating the effectiveness and complementarity of the proposed components.

In terms of model complexity and inference efficiency, Table 7 summarizes the number of parameters, GFLOPs, and inference time for each experiment. The baseline model in Experiment A contains 4.3 M parameters and 11.7 GFLOPs, with an inference time of 0.4 ms. Adding the LPConv module in Experiment B slightly increases GFLOPs to 12.1 and doubles inference time to 0.8 ms. Experiment C reduces parameters to 4.0 M and lowers GFLOPs to 11.1, maintaining moderate inference time. The addition of the IFM module in Experiment D raises parameters slightly to 4.4 M with a small increase in GFLOPs and inference time. Using SA-IoU loss in Experiment E does not affect complexity or speed. The combined modules in Experiments F to H keep parameters around 4.2 M, GFLOPs near 11.6, and inference times between 0.7 and 0.8 ms, demonstrating a good balance between accuracy and efficiency.

In order to thoroughly assess the effectiveness and superiority of the proposed method, several representative state-of-the-art object detection models, such as YOLOv5s, YOLOv10n, and SuperYOLO, are selected for comparison. The corresponding results are summarized in Table 8. The improved model RTMF-Net significantly outperforms the comparison methods in detection accuracy, achieving a mAP of 98.7%, which is the highest among all models. Meanwhile, RTMF-Net also demonstrates excellent performance in computational complexity, model size, and inference speed. It requires only 4.2 million parameters, 11.6 GFLOPs of computation, and achieves the lowest inference time of 0.8 milliseconds, which is substantially lower than other models. These results highlight its highly lightweight characteristic. In summary, RTMF-Net effectively reduces computational overhead while maintaining outstanding detection performance. Additionally, the mAP comparison curves are provided in Figure 7.

The generalization capability and practical applicability of the proposed model were assessed through comparative experiments conducted on the representative multimodal remote sensing dataset LLVIP, with the results summarized in Table 9. Although RTMF-Net demonstrates slightly lower detection accuracy than some more complex models, it shows significant advantages in model size and computational complexity. RTMF-Net possesses the lowest parameter count and computational cost among all compared models, greatly reducing dependence on hardware resources and exhibiting excellent lightweight characteristics and deployment of friend-lines. Compared with the best-performing model U2Fusion, RTMF-Net’s mAP is only about 1.0% lower, while its parameter count and computational load are substantially less than those of U2Fusion. This highlights RTMF-Net’s ability to maintain high detection performance while effectively controlling model complexity. These results further confirm RTMF-Net’s superiority in balancing accuracy and efficiency, demonstrating strong generalization ability and promising practical deployment potential.

Moreover, to further evaluate detection quality across different object sizes and detection thresholds, we adopt the COCO-style metrics on both the LLVIP and ODinMJ datasets.

On the LLVIP dataset, the model achieves an overall Average Precision (AP) of 56.7% under the IoU = 0.50:0.95 metric, with APs of 91.4% and 63.6% at IoU thresholds of 0.5 and 0.75, respectively. In terms of object sizes, the AP for large objects reaches 58.2%, while medium objects yield an AP of 22.0%. It is worth noting that the AP for small objects is reported as −1.0, which indicates that there are no small-sized objects in the validation set, making it impossible to compute a valid AP for this scale. The detailed results are presented in Table 10.

In contrast, the performance on the ODinMJ dataset demonstrates a substantial improvement across all metrics. The model achieves an overall AP of 69.0% and excels at standard IoU thresholds with 94.6% AP at IoU = 0.5 and 78.3% at IoU = 0.75. Notably, detection for small, medium, and large objects all exhibit significant gains, with APs of 56.6%, 72.7%, and 75.8%, respectively, demonstrating improved robustness in multi-scale object detection. The results on the ODinMJ dataset are presented in Table 11.

For recall, the LLVIP dataset yields an overall AR of 29.6% at maxDets = 1 and 63.1% at maxDets = 10 or 100. Meanwhile, the ODinMJ dataset presents a higher AR of 38.3% at maxDets = 1, and a consistent 72.9% at maxDets = 10 and 100. The ODinMJ recall performance is notably stronger across all object scales—AR values reach 61.6%, 76.3%, and 79.2% for small, medium, and large objects, respectively.

These results collectively demonstrate that the proposed method maintains superior scalability and robustness across datasets, particularly excelling in challenging multi-scale detection scenarios on ODinMJ.

To further validate the effectiveness of the proposed lightweight design, we conduct a comparative study with other representative lightweight backbones, namely MobileNetV3 and ShuffleNetV2. As shown in Table 12, our model has 4.2 million parameters, which is larger than ShuffleNetV2 with 0.64 million and MobileNetV3 with 0.23 million. However, its computational complexity remains moderate at 11.6 GFLOPs. In addition, the inference time of our model is 0.8 milliseconds, only slightly higher than ShuffleNetV2 at 0.6 milliseconds and MobileNetV3 at 0.4 milliseconds. These results demonstrate that the proposed design achieves a balanced trade-off between model size, computational cost, and detection performance, highlighting its suitability for real-time applications.

### 4.5. Visualization Results

A diverse set of four representative scenarios was selected from ODinMJ dataset to evaluate the effectiveness of the proposed improvements, and their visualization results are illustrated in Figure 8. Specifically, Figure 8a presents the detection results of the proposed multimodal detection model RTMF-Net, Figure 8b shows the results of Dual-YOLOv8n, Figure 8c illustrates the detection performance of the unimodal YOLOv8n on visible images, and Figure 8d provides the corresponding ground truth annotations. As shown, RTMF-Net achieves more accurate and stable detection across various challenging scenarios, without noticeable missed or false detections, significantly outperforming the baseline models. In the first and second groups representing near-ground occlusion scenes, YOLOv8n produces evident false positives, whereas RTMF-Net correctly identifies the occluded objects. In the third and fourth groups depicting low-light occlusion conditions, YOLOv8n fails to detect some objects, while RTMF-Net maintains consistent detection performance with high accuracy. These results demonstrate that RTMF-Net possesses superior robustness and detection capability under occlusion and poor lighting conditions, effectively validating the rationality and superiority of the proposed network architecture and fusion strategy.

Representative samples from the LLVIP dataset under diverse illumination conditions were selected for comparative analysis to examine the robustness of the proposed method, as depicted in Figure 9. In the figure, the first row presents the ground truth bounding boxes, the second row shows the detection results on the visible light modality, and the third row displays the results on the infrared modality. Specifically, Figure 9a illustrates the detection performance under sufficient daylight conditions, Figure 9b corresponds to a low-light environment at dusk, and Figure 9c–e, present typical nighttime scenarios with extremely low illumination. To address the challenges of object detection under such conditions, we specifically selected three representative sample images to evaluate the detection capability of the improved model. As shown, RTMF-Net consistently produces accurate detection results across various lighting conditions, maintaining high detection precision even in nighttime scenes without evident missed or false detections. This fully demonstrates the effectiveness of the improved model in multispectral fusion and its strong adaptability to complex environments.

To further verify the effectiveness of the proposed EFA module, we selected two representative scenes from the LLVIP dataset and visualized the attention maps using heatmaps. As shown in Figure 10. Figure 10a presents the original visible and infrared images, Figure 10b illustrates the attention heatmaps generated without the EFA module, and Figure 10c column displays the results after integrating EFA. It can be observed that the inclusion of EFA significantly enhances the model’s focus on object regions while reducing responses to background areas, demonstrating its effectiveness in improving object perception and suppressing background interference.

## 5. Discussion

Experimental results on the ODinMJ and LLVIP datasets demonstrate that RTMF-Net exhibits strong generalization capabilities and offers significant advantages in terms of computational efficiency. However, the improved model still presents certain limitations. Specifically, its performance may degrade in scenarios involving complex occlusions or extreme illumination variations, where the extracted features may lack sufficient discriminability, as shown in Figure 11. Additionally, while the model achieves a lightweight design, this may come at the cost of reduced representational capacity, potentially limiting its detection accuracy on highly diverse or small-scale targets.

In future work, to address the performance degradation of the model under extreme lighting conditions, we plan to enhance the model’s robustness by expanding the dataset with training data under various lighting conditions. Additionally, we plan to explore more advanced multi-modal fusion strategies to enhance the robustness of feature representations under challenging conditions. Furthermore, incorporating adaptive attention mechanisms and dynamic receptive field modulation may help the model better capture context-aware information. We will further complete the hardware deployment of the proposed model, aiming to efficiently integrate it into resource-constrained edge devices. This will allow us to validate its real-time performance and practical applicability, while achieving a balanced trade-off between detection accuracy and inference efficiency in real-world scenarios.

## 6. Conclusions

This study proposes a lightweight multimodal object detection network, RTMF-Net, which incorporates differentiated design and optimization of backbone structures tailored to the characteristics of different modality images. In the visible light branch, a CEFBlock composed of the C2f_FEM module and standard convolutions is introduced, effectively enhancing feature extraction capabilities for targets of varying scales. In the thermal infrared branch, the DLEBlock is constructed by combining improved pinwheel convolution, Laplacian operators, and the C2f structure, significantly improving the modeling of texture details and structural information in thermal infrared images. To fully leverage the complementarity between multimodal features, a weighted noise-reduction fusion module is designed which adaptively adjusts the contribution ratios of different modal features to suppress redundant background interference and enhance feature representation. Additionally, to address the morphological differences in similar targets caused by viewpoint variations in low-altitude UAV remote sensing, the shape-aware SA-IoU loss function is introduced to improve the robustness of target localization. Experiments conducted on two representative multimodal remote sensing datasets, ODinMJ and LLVIP, demonstrate that RTMF-Net achieves mAP improvements of 1.7% and 3.2%, respectively, with only 4.3 M parameters and 11.6 GFLOPs computation. The inference latency on the ODinMJ dataset is 1.4 ms, fully meeting real-time detection requirements. Future research will focus on further network compression and operator optimization, as well as exploring efficient deployment solutions on embedded platforms.

## Figures and Tables

**Figure 1 sensors-25-05631-f001:**
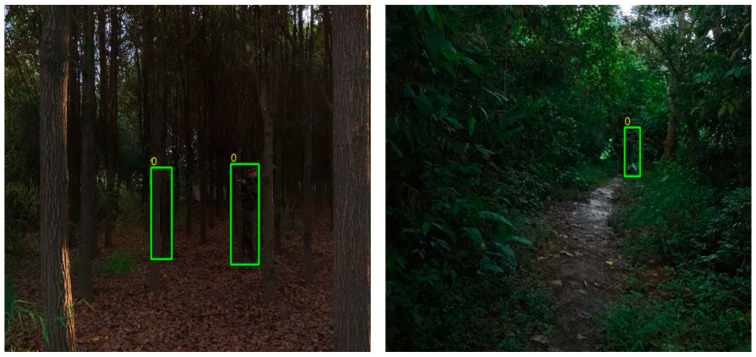
Representative image from the ODinMJ dataset: poor lighting and occlusion in dense forest lead to unclear target features and increased detection difficulty. The green square represents the object detection bounding box.

**Figure 2 sensors-25-05631-f002:**
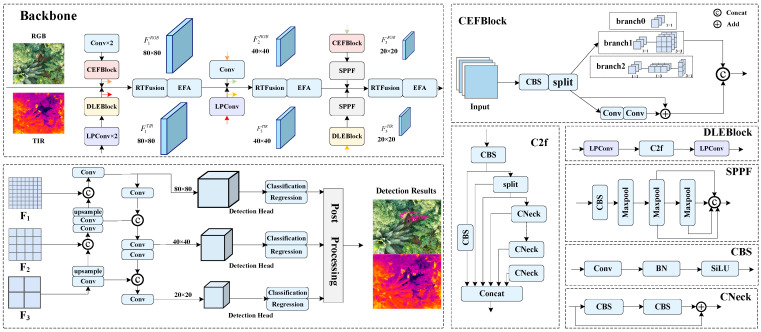
The structure of RTMF-Net.

**Figure 3 sensors-25-05631-f003:**
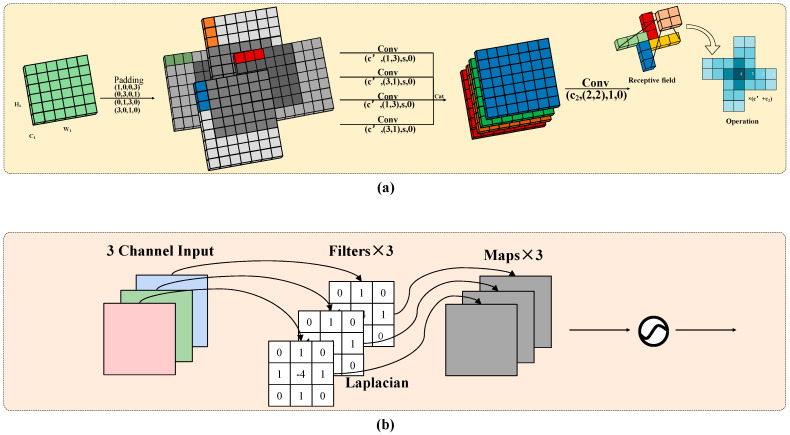
The structure of LPConv. (**a**) shows the structure of the PConv module and (**b**) illustrates the convolutional representation of the Laplacian.

**Figure 4 sensors-25-05631-f004:**
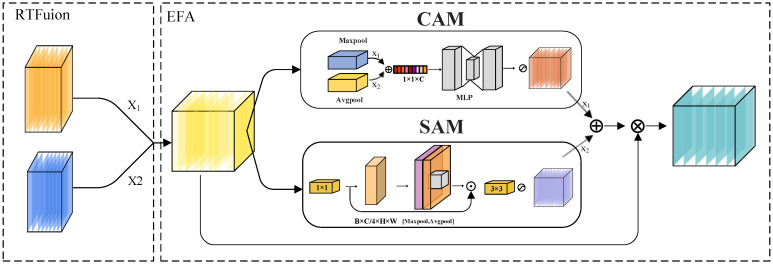
Weighted denoising module: dual-backbone features are weighted and enhanced via CAM and SAM to improve feature representation.

**Figure 5 sensors-25-05631-f005:**
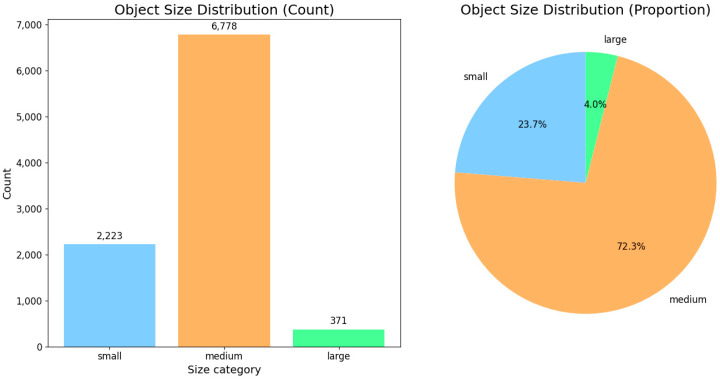
Category-wise object count and proportion statistics in the ODinMJ dataset.

**Figure 6 sensors-25-05631-f006:**
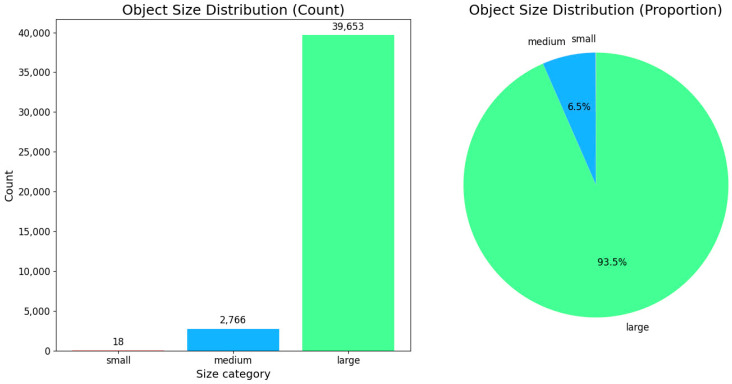
Category-wise object count and proportion statistics in the LLVIP dataset.

**Figure 7 sensors-25-05631-f007:**
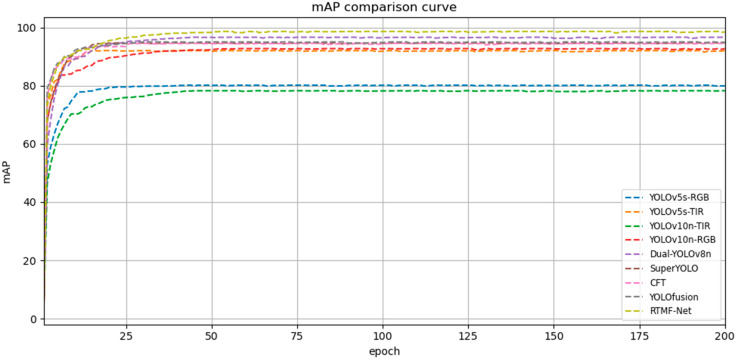
mAP comparison curve graph.

**Figure 8 sensors-25-05631-f008:**
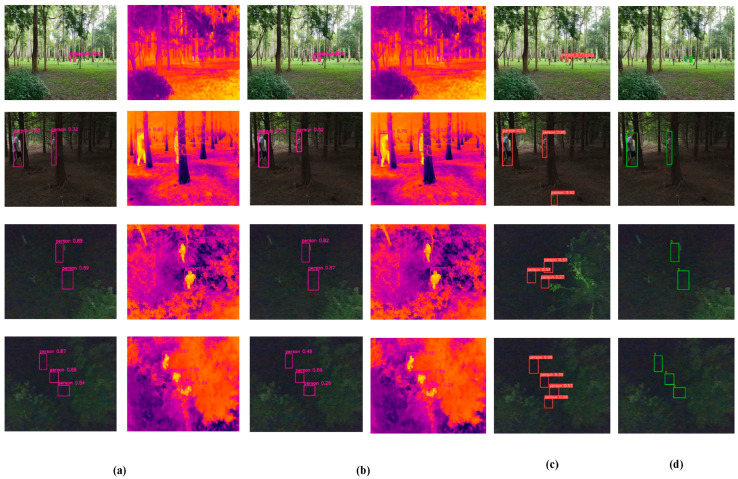
The detection results on the ODinMJ dataset are shown in (**a**–**d**), which, respectively, represent the results of the RTMF-Net, Dual-YOLOv8n, YOLOv8, and ground truth annotations.

**Figure 9 sensors-25-05631-f009:**
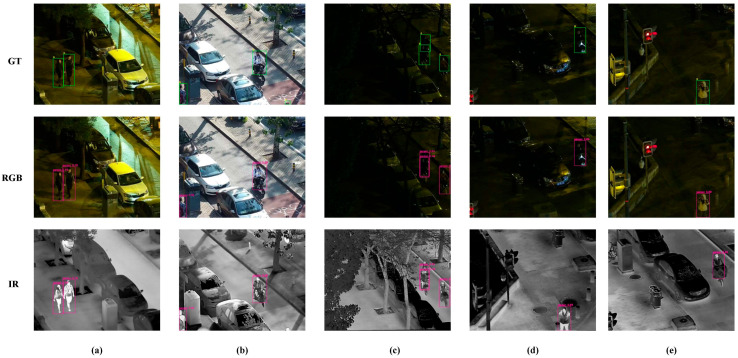
The detection results on the LLVIP dataset are shown in (**a**–**e**), which, respectively, correspond to detection outcomes under varying illumination conditions. The first row presents the ground truth bounding boxes, the second row shows the detection results based on RGB modality, and the third row displays the results based on IR modality.

**Figure 10 sensors-25-05631-f010:**
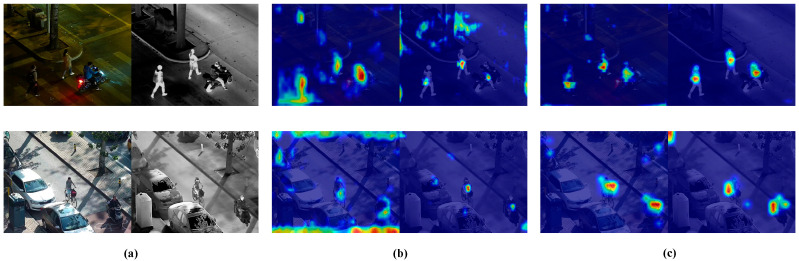
Visualization results of detection on the LLVIP dataset. (**a**) shows the original images, (**b**) displays the heatmaps without the EFA module, and (**c**) presents the heatmaps from the model with the EFA module integrated.

**Figure 11 sensors-25-05631-f011:**
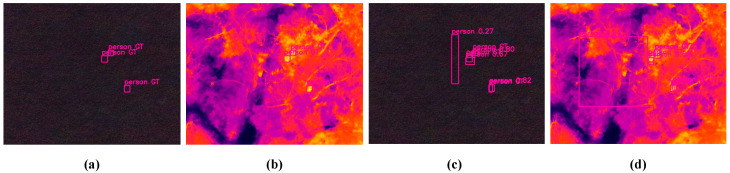
Detection results under scenarios involving complex occlusions or extreme illumination variations, where (**a,b**) correspond to the ground truth bounding boxes in the RGB and TIR images, respectively, and (**c**,**d**) show the predicted detection boxes in the RGB and TIR images, respectively.

**Table 1 sensors-25-05631-t001:** Object statistics of the ODinMJ dataset.

Scale	Number	Proportion
Small	2223	23.72%
Middle	6778	72.32%
Large	371	3.96%

**Table 2 sensors-25-05631-t002:** Object statistics of the LLVIP dataset.

Scale	Number	Proportion
Small	18	0.04%
Middle	2766	6.52%
Large	39,653	93.44%

**Table 3 sensors-25-05631-t003:** Ablation study on the Alpha parameter of SA-IoU.

Setting	P (%)	R (%)	mAP_50_ (%)	mAP_50–95_ (%)
(0.5, 0.5)	**94.3**	**89.1**	**94.9**	**59.5**
(0.6, 0.4)	93.9	88.7	94.7	58.7
(0.4, 0.6)	94.8	88.9	94.8	58.8
(0.7, 0.3)	93.6	87.3	94.4	58.5
(0.3, 0.7)	95.3	86.7	94.5	59.4

**Table 4 sensors-25-05631-t004:** Ablation study on the P, R, mAP_50_, and mAP_50–95_ for the ODinMJ dataset.

Model	LPConv	C2f_FEM	IFM	SA-IoU	P (%)	R (%)	mAP_50_ (%)	mAP_50–95_ (%)
A					95.4	92.5	96.7	69.9
B	√				96.4	94.5	97.8	72.7
C		√			96.2	94.5	97.7	72.8
D			√		96.4	94.8	97.9	73.1
E				√	96.1	93.3	97.1	71.1
F	√	√			97.0	95.2	98.0	73.9
G	√	√	√		97.2	95.6	98.3	75.0
H	√	√	√	√	**97.4**	**96.3**	**98.7**	**75.3**

**Table 5 sensors-25-05631-t005:** Ablation study on the Params, GFLOPs, and IT for the ODinMJ dataset.

Model	LPConv	C2f_FEM	IFM	SA-IoU	Params (M)	GFLOPs	IT (ms)
A					4.3	11.7	**0.5**
B	√				4.3	12.1	0.7
C		√			4.0	11.1	0.6
D			√		4.4	11.7	**0.5**
E				√	4.3	11.7	**0.5**
F	√	√			4.2	11.6	0.8
G	√	√	√		4.2	11.6	0.9
H	√	√	√	√	4.2	11.6	0.9

**Table 6 sensors-25-05631-t006:** Ablation study on the P, R, mAP_50_, and mAP_50–95_ for the LLVIP dataset.

Model	LPConv	C2f_FEM	IFM	SA-IoU	P(%)	R(%)	mAP_50_ (%)	mAP_50–95_ (%)
A					93.9	87.6	94.4	57.8
B	√				94.1	89.2	94.9	60.1
C		√			95.0	89.1	95.1	60.3
D			√		94.4	89.6	95.2	60.5
E				√	94.2	88.1	94.8	59.5
F	√	√			95.1	90.2	95.2	61.2
G	√	√	√		**95.5**	90.8	95.5	61.4
H	√	√	√	√	95.3	**91.3**	**95.7**	**61.3**

**Table 7 sensors-25-05631-t007:** Ablation study on the Params, GFLOPs, and IT for the LLVIP dataset.

Model	LPConv	C2f_FEM	IFM	SA-IoU	Params (M)	Gflops	IT (ms)
A					4.3	11.7	**0.4**
B	√				4.3	12.1	0.8
C		√			4.0	11.1	0.7
D			√		4.4	11.7	0.7
E				√	4.3	11.7	**0.4**
F	√	√			4.2	11.6	0.7
G	√	√	√		4.2	11.6	0.8
H	√	√	√	√	4.2	11.6	0.8

**Table 8 sensors-25-05631-t008:** Comparison experiments on the ODinMJ dataset.

Model	Data	Parameter (M)	Gflops	mAP_50_ (%)	IT (ms)
YOLOv5s	RGB	7.2	14.9	80.2%	1.9
YOLOv5s	TIR	7.2	14.7	92.1%	1.9
YOLOv10n	RGB	8.2	**2.7**	78.3%	1.7
YOLOv10n	TIR	8.2	**2.7**	92.8%	1.7
Dual-YOLOv8n	RGB+TIR	4.3	11.7	96.7%	0.9
SuperYOLO [15]	RGB+TIR	4.9	56.3	94.8%	19.4
CFT [8]	RGB+TIR	44.9	17.9	94.6%	62.5
YOLOfusion [9]	RGB+TIR	24.9	35.7	95.1%	55.6
RTMF-Net	RGB+TIR	**4.2**	11.6	**98.7%**	**0.8**

**Table 9 sensors-25-05631-t009:** Comparison experiments on the LLVIP dataset.

Model	Data	Parameter (M)	Gflops	mAP_50_ (%)
Dual-YOLOv8n	RGB+IR	4.3	**11.4**	92.6%
SuperYOLO [15]	RGB+IR	4.9	56.3	93.2%
CFT [8]	RGB+IR	44.9	17.9	94.6%
MOD-YOLO [44]	RGB+IR	24.9	35.7	95.2%
Infusion-Net [45]	RGB+IR	239.2	241.8	98.6%
MMI-Det [46]	RGB+IR	207.6	222.9	**98.9%**
FQDNet_n [40]	RGB+IR	4.7	16.9	95.5%
U2Fusion [47]	RGB+IR	47.7	1032.6	96.7%
IFCNN [48]	RGB+IR	47.1	206.2	95.5%
DenseFuse [49]	RGB+IR	47.1	238.1	95.8%
RTMF-Net	RGB+IR	**4.2**	11.6	95.7%

**Table 10 sensors-25-05631-t010:** Detection evaluation on ODinMJ dataset (COCO metrics).

IoU Threshold	Area	MaxDets	Value (%)
AP 0.50:0.95	all	100	69.0
AP 0.5	all	100	94.6
AP 0.75	all	100	78.3
AP 0.50:0.95	small	100	56.6
AP 0.50:0.95	medium	100	72.7
AP 0.50:0.95	large	100	75.8
AR 0.50:0.95	all	1	38.3
AR 0.5	all	10	72.9
AR 0.75	all	100	72.9
AR 0.50:0.95	small	100	61.6
AR 0.50:0.95	medium	100	76.3
AR 0.50:0.95	large	100	79.2

**Table 11 sensors-25-05631-t011:** Detection evaluation on LLVIP dataset (COCO metrics).

IoU Threshold	Area	MaxDets	Value (%)
AP 0.50:0.95	all	100	56.7
AP 0.5	all	100	91.4
AP 0.75	all	100	63.6
AP 0.50:0.95	small	100	−1
AP 0.50:0.95	medium	100	22.0
AP 0.50:0.95	large	100	58.2
AR 0.50:0.95	all	1	29.6
AR 0.5	all	10	63.1
AR 0.75	all	100	63.1
AR 0.50:0.95	small	100	−1
AR 0.50:0.95	medium	100	34.1
AR 0.50:0.95	large	100	64.6

**Table 12 sensors-25-05631-t012:** Compared with other lightweight backbones.

Backbone	Parameter (M)	Gflops	IT (ms)
ShufflenetV2	0.64	16.5	0.6
MobilenetV3	**0.23**	**5.8**	**0.4**
Ours	4.2	11.6	0.8

## Data Availability

The original contributions presented in this study are included in the article. Further inquiries can be directed to the corresponding author.

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
