# Peer review of "RTMF-Net: A Dual-Modal Feature-Aware Fusion Network for Dense Forest Object Detection"

_sensors, 2025, doi:10.3390/s25185631_

Round 1
Reviewer 1 Report
Comments and Suggestions for Authors
1.The paper emphasizes the lightweight design of RTMF-Net, which is suitable for real-time applications, but does not provide a comparison of training and testing time with other models.
2. It is suggested to supplement the training accuracy and loss curves of the proposed methods and comparison methods
3. The manuscript presents a large amount of data in the form of tables, but all variables in the table do not have units, and the variable names are too short and not explained, resulting in poor readability. It is recommended to supplement them. ​
4. In Table 9, the GFLOPs of U2Fusion (1032.6) were significantly higher than those of RTMF Net (11.6), but the reason for the difference was not explained. Suggest supplementing the reasons for differences and conducting comparative experiments to highlight the advantages.
5.In the conclusions, the author mentions that the performance of the model decreases under extreme lighting conditions, but does not provide any direction for improvement.
OK
Author Response
Comments 1: [The paper emphasizes the lightweight design of RTMF-Net, which is suitable for real-time applications, but does not provide a comparison of training and testing time with other models.]
Response 1: Thank the experts for valuable advice. We have included inference time comparisons in our experiments on the ODinMJ dataset. Since some of the code has not been open-sourced, we only report inference time on ODinMJ. We believe this is sufficient to demonstrate that our model can be applied in real-time scenarios.
This modification is on page 17, line 616.
Comments 2: [It is suggested to supplement the training accuracy and loss curves of the proposed methods and comparison methods.]
Response 2: Thank the experts for valuable advice. We fully agree that training accuracy and loss curves can provide deeper insights into the model optimization process. We have added some comparative accuracy curves on the ODinMJ dataset which is shown in Figure 8. Since different models have different loss outputs, they cannot be directly compared together. Additionally, as the code for some models is not publicly available, we can only provide accuracy comparison curves for the ODinMJ dataset.
This modification is on page 17, line 616.
Comments 3:[The manuscript presents a large amount of data in the form of tables, but all variables in the table do not have units, and the variable names are too short and not explained, resulting in poor readability. It is recommended to supplement them.]
Response 3: Thank the experts for valuable advice. We have added units to the parameters of each table.
These improvements can be found on lines 462 and 465 on page 12, lines 512, 517, and 518 on page 14, lines 561 and 562 on page 15, lines 616 and 632 on page 17, lines 634 and 633 on page 19, and lines 669 on page 19.
Comments 4: [In Table 9, the GFLOPs of U2Fusion (1032.6) were significantly higher than those of RTMF Net (11.6), but the reason for the difference was not explained. Suggest supplementing the reasons for differences and conducting comparative experiments to highlight the advantages.]
Response 4: Thank the experts for valuable advice, The large GFLOPs difference mainly results from architectural design: U2Fusion employs a deep encoder–decoder structure with dense connections and pixel-level fusion operations, while RTMF-Net adopts lightweight asymmetric convolutions and task-specific feature extraction. GFLOPs are calculated directly from the model architecture and do not require additional experiments. To highlight the advantage, we have emphasized in the revised manuscript the trade-off between accuracy and computational cost, where RTMF-Net achieves competitive mAP while reducing GFLOPs by nearly two orders of magnitude. This demonstrates its superiority in real-time deployment scenarios without the need for further experiments, If you think other experiments are needed, please explain to us in detail.
Comments 5: [In the conclusions, the author mentions that the performance of the model decreases under extreme lighting conditions, but does not provide any direction for improvement.]
Response 5: Thank the experts for valuable advice, We have added future improvement plans.
This modification is on page 22, lines 739 to 759.
Reviewer 2 Report
Comments and Suggestions for Authors
Review of the manuscript
“RTMF-Net: A Dual-Modal Feature-Aware Fusion Network for Dense Forest Object Detection” by Xiaotan Wei , Zhensong Li, Yutong Wang, and Shiliang Zhu
The paper presents the RTMF-Net method for detecting objects in a dense forest based on RGR and TIR images. A distinctive feature of this method is that it does not require significant computing power.
- An obvious advantage is the superiority of the method by the mAP criterion, which reaches 98.7% on the ODinMJ dataset. However, its closest competitors (see Abstract and Table 8, Dual-YOLOv8n) have mAP=96.7%, i.e. the difference is 2%. As for the LLVIP dataset, for the proposed method mAP=95.7%, and for MMI-Det mAP=98.9%, i.e. the proposed method is worse by 3.2%. The reader is interested in the answer to the question: is the difference of 2% or 3.2% significant to claim that the proposed method is significantly superior to other methods?
- When discussing the results, examples should be given that explain in which situations an increase of 2% or a decrease of 3.2% will not allow detecting objects in a dense forest based on RGR and TIR images.
- The Abstract provides the value mAP=96.0%. The text of the article does not mention this value (maybe mAP=95.7%?). The value mAP=96.0% should be clarified.
- The manuscript requires minor editorial revision, e.g., “Xiaotan Wei, Zhensong Li and Yutong Wang, and Shiliang Zhu”.
The article may be published in the journal Sensors after taking these comments into account.
Author Response
Comments 1: [An obvious advantage is the superiority of the method by the mAP criterion, which reaches 98.7% on the ODinMJ dataset. However, its closest competitors (see Abstract and Table 8, Dual-YOLOv8n) have mAP=96.7%, i.e. the difference is 2%. As for the LLVIP dataset, for the proposed method mAP=95.7%, and for MMI-Det mAP=98.9%, i.e. the proposed method is worse by 3.2%. The reader is interested in the answer to the question: is the difference of 2% or 3.2% significant to claim that the proposed method is significantly superior to other methods.]
Response 1: Thank the experts for valuable advice. On the ODinMJ dataset, our method achieves an mAP of 98.7%, which is 2% higher than the closest competitor, Dual-YOLOv8n, while maintaining a lightweight design. It is worth emphasizing that when the mAP has already exceeded 95%, achieving further improvements becomes increasingly difficult. Therefore, even a 1–2% gain at this level is practically meaningful and demonstrates the robustness of our approach.On the LLVIP dataset, our method obtains an mAP of 95.7%, which is slightly lower than MMI-Det (98.9%). However, MMI-Det comes with a substantial computational cost (207.6M parameters and 222.9 GFLOPs), whereas our method only requires 4.7M parameters and 11.6 GFLOPs. Despite the significantly lower complexity, our approach still achieves a competitive accuracy, highlighting a favorable trade-off between efficiency and performance, which is crucial for real-world deployment.
Comments 2: [When discussing the results, examples should be given that explain in which situations an increase of 2% or a decrease of 3.2% will not allow detecting objects in a dense forest based on RGR and TIR images.]
Response 2: Thank the experts for valuable advice. We conducted comparative visualization experiments, as shown in Figure 7, where we compared the detection results of Dual-YOLOv8 and RTMF-Net, illustrating how the improvement in accuracy translates into differences in detection performance.Perhaps we have misunderstood your comment. If you need any other proof, please explain it in detail.
Comments 3: [The Abstract provides the value mAP=96.0%. The text of the article does not mention this value (maybe mAP=95.7%?). The value mAP=96.0% should be clarified.]
Response 3: Thank the experts for valuable advice. This was an error caused by our carelessness and has been corrected.
This modification is in line 32 of the abstract.
Comments 4: [The manuscript requires minor editorial revision, e.g., “Xiaotan Wei, Zhensong Li and Yutong Wang, and Shiliang Zhu”]
Response 4:Thank the experts for valuable problem. We have fixed the issue.
This modification is on the 4th line of the homepage.
Reviewer 3 Report
Comments and Suggestions for Authors
- The paper presents RTMF-Net as a novel solution, but the core architecture follows traditional dual-stream patterns established in existing multimodal detection frameworks. The claimed novelty appears to be primarily in component-level modifications rather than architectural innovation.
- The related work section also indicates that the proposed approach largely recombines existing techniques without addressing fundamental limitations of current multimodal fusion methods. The paper did not establish a clear gap in existing solutions or explain why the specific combination of techniques chosen here is necessary beyond incremental performance gains.
- In lines 252-282, while the three-branch structure combines asymmetric convolutions and multi-directional modeling, this approach has been extensively explored in prior work. The mathematical formulation in Equations (1)-(4) represents standard operations without novel theoretical insights. The claim of handling "geometric distortions" in remote sensing is not substantiated with appropriate analysis.
- In lines 290-327, why did the authors combine Pinwheel Convolution with Laplacian edge detection, what's the rationale? The paper doesn't demonstrate why this specific combination outperforms existing feature extraction methods.
- In ablation studies, it seems the individual module contributions are modest (typically 1-2% mAP improvement), and the hyperparameter sensitivity analysis is superficial. The fact that different parameter combinations yield similar results suggests the proposed components may not be addressing distinct problems, and raises questions for the necessity of the multi-component approach.
- In lines 594-617, comparing models trained on different modality combinations without proper cross-validation makes the results difficult to interpret. The paper doesn't address whether the improvements come from the proposed architecture or simply from using more training data (multimodal vs unimodal).
- In lines 621-646, the substantial performance difference between ODinMJ (69.0% AP) and LLVIP (56.7% AP) suggests the model may be overfitted to specific dataset characteristics rather than learning generalizable multimodal representations. The -1.0 AP for small objects on LLVIP indicates inadequate validation data.
Author Response
Comments 1: [The paper presents RTMF-Net as a novel solution, but the core architecture follows traditional dual-stream patterns established in existing multimodal detection frameworks. The claimed novelty appears to be primarily in component-level modifications rather than architectural innovation.]
Response 1: Thank the experts for valuable advice. We acknowledge that the overall dual-stream paradigm has been widely adopted in multimodal detection. Our motivation was not to redesign the entire architecture from scratch but to address the practical limitations of existing frameworks within this paradigm. Specifically, we introduced novel component-level modules (e.g., asymmetric convolutions, pinwheel–Laplacian fusion) that enhance geometric robustness and edge-preserving feature extraction, while keeping the architecture lightweight and deployment-friendly. As demonstrated in our experiments, these targeted innovations bring consistent and notable improvements across multiple datasets. Thus, our contribution lies in advancing the effectiveness of the dual-stream framework through carefully designed and practically motivated modules.
Comments 2: [The related work section also indicates that the proposed approach largely recombines existing techniques without addressing fundamental limitations of current multimodal fusion methods. The paper did not establish a clear gap in existing solutions or explain why the specific combination of techniques chosen here is necessary beyond incremental performance gains.]
Response 2: Thank the experts for valuable advice. We sincerely thank the reviewer for this insightful comment. While our work indeed builds upon the dual-stream paradigm, our contribution is not a simple recombination of existing methods. Instead, we specifically designed the proposed modules to address two key limitations of current multimodal fusion frameworks: (1) inadequate preservation of modality-specific features, which we mitigate through CEFBlock and DLEBlock tailored for RGB and TIR respectively; and (2) redundancy and imbalance in cross-modal fusion, which we alleviate with the Weighted Denoising Fusion Module and Enhanced Fusion Attention. These components were carefully chosen and integrated based on practical limitations of prior approaches, rather than arbitrarily combined. As demonstrated in our experiments, this targeted design achieves consistent improvements in both accuracy and efficiency. To make this clearer, we have revised the contributions section to explicitly highlight the research gap and the rationale behind our module choices.
This modification is on page 4, lines 127 to 135.
Comments 3: [In lines 252-282, while the three-branch structure combines asymmetric convolutions and multi-directional modeling, this approach has been extensively explored in prior work. The mathematical formulation in Equations (1)-(4) represents standard operations without novel theoretical insights. The claim of handling "geometric distortions" in remote sensing is not substantiated with appropriate analysis.]
Response 3: Thank the experts for valuable advice. We have provided a more detailed explanation of geometric distortions.
This modification is on page 7, lines 258 to 26.
Comments 4:[In lines 290-327, why did the authors combine Pinwheel Convolution with Laplacian edge detection, what's the rationale? The paper doesn't demonstrate why this specific combination outperforms existing feature extraction methods.]
Response 4: Thank the experts for valuable advice. We provided a more detailed explanation of why PConv and Laplace operators are combined.
This modification is on page 8, line 327 to page 9, line 340.
Comments 5: [In ablation studies, it seems the individual module contributions are modest (typically 1-2% mAP improvement), and the hyperparameter sensitivity analysis is superficial. The fact that different parameter combinations yield similar results suggests the proposed components may not be addressing distinct problems, and raises questions for the necessity of the multi-component approach.]
Response 5: Thank the experts for valuable advice. In RTMF-Net, we designed dedicated feature extraction modules for the two different modality backbones and further introduced a weighted fusion module. We acknowledge that, in the ablation study, the performance improvement of each individual module appears relatively modest (around 1–2% mAP), which may be partly due to the already high baseline performance. Nevertheless, we believe these improvements are still meaningful. More importantly, the modules are complementary in functionality, and when working together, they lead to a considerably greater improvement than when used individually, enabling RTMF-Net to achieve the best detection accuracy. Therefore, we view these modules not as redundant, but as effective components that enhance multimodal feature representation through their synergistic interaction.
Comments 6:[Thank the experts for valuable advice. In lines 594-617, comparing models trained on different modality combinations without proper cross-validation makes the results difficult to interpret. The paper doesn't address whether the improvements come from the proposed architecture or simply from using more training data (multimodal vs unimodal).]
Response 6: Thank the experts for valuable advice. All experiments in this work were conducted on the official splits provided by the dataset authors, and therefore we did not perform additional cross-validation. In the comparative experiments, we evaluated both unimodal and multimodal models, as shown in Table 8. We believe this demonstrates that the accuracy improvements are not merely due to the increase in data volume.
Comments 7: [In lines 621-646, the substantial performance difference between ODinMJ (69.0% AP) and LLVIP (56.7% AP) suggests the model may be overfitted to specific dataset characteristics rather than learning generalizable multimodal representations. The -1.0 AP for small objects on LLVIP indicates inadequate validation data.]
Response 7: Thank the experts for valuable advice. Since currently available RGB–TIR datasets present certain limitations, we adopted LLVIP as one of our evaluation benchmarks. Following COCO-style statistics, we found that LLVIP indeed lacks small-object instances; however, it shares challenges such as occlusion with ODinMJ, making it still valuable for validating RTMF-Net. We would be glad to incorporate additional experiments on other datasets if the reviewer could kindly suggest more suitable benchmarks.
Reviewer 4 Report
Comments and Suggestions for Authors
summary and the contributiosns
The paper introduces RTMF-Net, a lightweight dual-stream network for RGB–TIR remote sensing object detection that employs modality-specific enhancement modules (CEFBlock for RGB, DLEBlock for TIR) to improve semantic and structural feature extraction. It further proposes a Weighted Denoising Fusion Module with Enhanced Fusion Attention to suppress redundancy and emphasize salient regions, along with a Shape-Aware IoU loss for more robust localization. It shows good performance on the studied dataset.
Major Weaknesses:
- The novelty of the manuscrip is limmited, While the backbone modules (CEFBlock, DLEBlock) and fusion module are interesting, they resemble incremental modifications of existing enhancement and attention-based strategies.
- Reported mAP scores (up to 98.7%) are extremely high compared to most multimodal detection literature, raising concerns about overfitting, dataset difficulty, or lack of cross-dataset validation. It would be good if the authors conduct more ablation experiments and check the above comment. How the model work on small object, what is the mAP @ 50-95% ? What is the recall?
- SA-IoU is conceptually useful, but aspect ratio–aware IoU variants already exist in the literature (e.g., GIoU, DIoU, CIoU).
Comments:
- In the related work secion, it would be good if the authors also provide literature about the transformer based models like DETR, RFDETR, etc. And also the very recent methods like vision language based modlels? Do you thinks the VLM models can work on the task in this paper.
In the methodology section :
- Details are needed in the Caption of figure 2, especially the key items in the figures, what are the main contributions of this paper and what are the baseline.
- What is the rationale behind of using these modules? The motivation also should be clear in the contributions paragraph.
- What happen if the proposed modules be applied on other baseline detectors like YOLO, DETR,etc. It would be good if the authors show some experiments on this.
Resutls section
- In 4.1, the details like python, pytorch are good to be reported on the github page of the model, Do the authors intend to release the code and dataset?
- In 4.1, it is better to use the present tense instead of past.
- Move the datset in a separate subsection, e.g., dataset, in 4.1.
- Some hyperparams are missing in the implementation details of 4.1, the augmentation technique, how the model trained, the pre-training, fine-tuning details are needed here; how long did it take to train the model;what optimizer was used.
- It would be good if the authors bold the best performances in the tables and provide more details in the captions. It help reader to better understand the resutls.
- The paper highlights efficiency, but comparisons with other lightweight backbones (e.g., MobileNetV3, ShuffleNet, GhostNet) are missing. Without these baselines, the efficiency claim is less convincing.
- What is the efficiency of the model in terms of Frames-per-second? Performance comparisons are primarily mAP-based. Latency measurements (FPS on real hardware) would better support real-time claims.
- Show some qualitative results on the limitaions of seciton 5.
Author Response
We sincerely thank the reviewer for the valuable comments. In our work, we have already reported the mAP@50-95 results, which are jointly determined by precision (P) and recall (R). Both P and R have been included in the COCO evaluation metrics as well as in the ablation experiments conducted on the two datasets. Moreover, the COCO evaluation protocol also provides performance evaluation across different object scales.
Comments 1: [In the related work section, it would be good if the authors also provide literature about the transformer based models like DETR, RFDETR, etc. And also the very recent methods like vision language based models? Do you thinks the VLM models can work on the task in this paper.]
Response 1: Thank the experts for valuable advice. We have added Transformer-based detection methods. Regarding the Vision-Language Models (VLMs) mentioned by the reviewer, their main focus lies in cross-modal alignment and reasoning between images and natural language, while our work concentrates on feature fusion and object detection across RGB and TIR visual modalities. Thus, the research scopes are different. Nevertheless, we believe that certain cross-modal modeling ideas from VLMs (e.g., multimodal attention mechanisms) could inspire future studies on more complex multimodal detection tasks.
This modification is on page 4, line 173 to page 5, line 181.
Comments 2: [Details are needed in the Caption of figure 2, especially the key items in the figures, what are the main contributions of this paper and what are the baseline.What is the rationale behind of using these modules? The motivation also should be clear in the contributions paragraph.]
Response 2: Thank the experts for valuable advice. We have annotated the operation in Figure 2, and our model is an improvement based on YOLOv8. We have added a more detailed description in the contribution paragraph.
This modification is on page 4, lines 127 to 135.
Comments 3: [What happen if the proposed modules be applied on other baseline detectors like YOLO, DETR,etc. It would be good if the authors show some experiments on this.]
Response 3: Thank the experts for valuable advice. Our proposed RTMF-Net is an improvement based on YOLOv8. As for the DETR series, we consider their Transformer-based design to involve relatively high computational overhead, which does not align with our lightweight design objective.
Comments 4: [In 4.1, the details like python, pytorch are good to be reported on the github page of the model, Do the authors intend to release the code and dataset? ]
Response 4: Thank the experts for valuable advice. We have uploaded the code on Figshare.
Comments 5: [In 4.1, it is better to use the present tense instead of past.]
Response 5: Thank the experts for valuable advice. We have revised it to the present tense.
This modification is on page 11 lines 442 to 449.
Comments 6: [Move the datset in a separate subsection, e.g., dataset, in 4.1. ]
Response 6: Thank the experts for valuable advice. We have presented the DATASET section in a separate section.
This modification is on page 11 lines 450.
Comments 7: [Some hyperparams are missing in the implementation details of 4.1, the augmentation technique, how the model trained, the pre-training, fine-tuning details are needed here; how long did it take to train the model;what optimizer was used. ]
Response 7: Thank the experts for valuable advice. We have added more detailed parameter settings.However, the training time may vary greatly depending on different devices, so we only explained the training epochs.
This modification is on page 11 lines 450.
Comments 8:[It would be good if the authors bold the best performances in the tables and provide more details in the captions. It help reader to better understand the resutls. ]
Response 8: Thank the experts for valuable advice. We have bolded the best values in the tables and indicated this in the text.
These improvements can be found on lines 462 and 465 on page 12, lines 512, 517, and 518 on page 14, lines 561 and 562 on page 15, lines 616 and 632 on page 17, lines 634 and 633 on page 19, and lines 669 on page 19.
Comments 9: [The paper highlights efficiency, but comparisons with other lightweight backbones (e.g., MobileNetV3, ShuffleNet, GhostNet) are missing. Without these baselines, the efficiency claim is less convincing. ]
Response 9: Thank the experts for valuable advice. We have added comparative experiments using different backbones, evaluating inference time, number of parameters, and computational cost to demonstrate the lightweight performance of the improved model.
This modification is on page 19 lines 669.
Comments 10: [What is the efficiency of the model in terms of Frames-per-second? Performance comparisons are primarily mAP-based. Latency measurements (FPS on real hardware) would better support real-time claims. ]
Response 10: Thank the experts for valuable advice. We have included inference time in the ablation studies, which is the reciprocal of FPS and can serve as a measure of model efficiency. Similarly, inference time is also added in the comparative experiments with different backbones to evaluate the efficiency of the models.
This modification is on page 17 lines 616.
Comments 11:[ Show some qualitative results on the limitaions of seciton 5.]
Response 11: Thank the experts for valuable advice. We are not entirely sure what the reviewer specifically means by “qualitative results.” In our visualization experiments, we have already presented comparisons of the detection results between the improved model and the baseline model. If the reviewer expects other types of qualitative results, please specify, and we will include them in a subsequent revision.
Round 2
Reviewer 1 Report
Comments and Suggestions for Authors
1.There are some formatting issues with tables and images, such as incorrect positioning.
2.Suggest changing the curves in the epoch-mAP figure to a dashed line.
3.Carefully check the English and format before publication.
Comments on the Quality of English LanguageOK
Author Response
Comments 1: There are some formatting issues with tables and images, such as incorrect positioning.
Response 1: Thank the experts for valuable advice. We have made revisions to address the issues that may have arisen during the upload of the paper, and have now made changes to the format.
Comments 2: Suggest changing the curves in the epoch-mAP figure to a dashed line.
Response 2: Thank the experts for valuable advice. We have replaced it with a dotted line.
Comments 3: Carefully check the English and format before publication.
Response 3: Thank the experts for valuable advice. We have carefully checked and improved the English language and formatting throughout the manuscript.
Reviewer 4 Report
Comments and Suggestions for Authors
The authors addressed my previous comments, only the following minor comments need attention by the authors:
1- Check and correct the Table in line 582-584 page 16. Please also check and revise the Table and Figures in pages 17 and 18.
2- For Comments 11, the query was to provide some failure case where the model fail. show somoe example images.
Author Response
Comments 1: Check and correct the Table in line 582-584 page 16. Please also check and revise the Table and Figures in pages 17 and 18.
Response 1: Thank the experts for valuable advice. We have made revisions to the format, which may have been an error during the upload of the paper.
Comments 2: For Comments 11, the query was to provide some failure case where the model fail. show somoe example images.
Response 2: Thank the experts for valuable advice. We have added some examples of failed detections.
This modification is on page 21, line 720 to page 22, line 742.